# Modular Structures of Trade Flow Networks in International Commodities

Zannatul Mawa Koli [1], Ashadun Nobi [1], Mahmudul Islam Rakib [1], Jahidul Alam [1] and Jae Woo Lee [2,*]

1   Department of Computer Science and Telecommunication Engineering, Noakhali Science and Technology University, Noakhali 3814, Bangladesh; ashadunnobi_305@yahoo.com (A.N.); rakibmahmud.inbox@gmail.com (M.I.R.); jahidsagar@gmail.com (J.A.)
2   Department of Physics, Inha University, Incheon 22212, Republic of Korea
*   Correspondence: jaewlee@inha.ac.kr; Tel.: +82-32-860-7660

**Abstract:** We explore the evolution of modular structure within the International Trade Network (ITN) for eight commodities, employing the Louvain module optimization method. The interactions among countries in the realm of trade are shaped by various factors, including economic conditions and geographical proximity. These countries are often categorized into continental groups, a classification that frequently persists even after the detecting process of modules. Nonetheless, African countries display a penchant for shifting among different modules over time. Observations of module trends unveil the increase in regional trade up until 2005, followed by plateaus marked with interruptions during significant crises, such as the 2012–2014 EU recession and the 2018 trade war. Notably, the 2018 trade war witnessed a sharp upsurge in module, attributed to robust alliances between major players like China and the USA. These modular dynamics are not uniform across different commodities; they exhibit varying degrees of module and distinct responses during times of crisis, with human-made goods displaying heightened sensitivity. Core nations, such as the USA, Germany, China, and Japan, exert significant influence over the commodities and often demonstrate a cohesive approach when navigating through crises. The analysis of modular dynamics provides valuable insights into global trade trends, fostering sustainability in trade practices, and comprehending the impacts of crises on various commodities.

**Keywords:** trade network; commodity; module; complex network

## 1. Introduction

Every country maintains interconnected trade relationships with one another, relying on the global value chain to procure the necessary components for producing goods [1]. This interdependence has led to the creation of a complex and intricate international trade network (ITN), also referred to as the world trade web (WTW) [2]. In this network, countries serve as nodes, representing entities involved in trading goods, while the links signify the import–export relationships between them. Within this complex network, nodes often form modular structures due to their interactions. These modules, or communities, exhibit dense internal connections, indicating strong ties among the nodes within each group. Detecting and understanding these modules allows us to extract valuable information about their characteristics and sheds light on the overall structure of the network [3].

In large networks like the ITN, not all nodes and links play equal roles. Certain nodes hold significant importance in facilitating the distribution of traffic flow, much like key intersections in a traffic network. Similarly, specific links carry a substantial share of the traffic flow because they connect crucial groups of nodes that play vital roles within the overall traffic network [4]. Numerous methods have been employed to identify modular structures in complex networks, as evidenced by a range of studies [5–19]. For instance, De Leo et al. utilized the module core detection method in transportation networks to pinpoint significant nodes [4]. On the other hand, Zhu et al. applied module optimization and

core detection techniques to observe the rise of China in the International Trade Network (ITN) and identified the core of the Asia-Oceania module [5]. Moreover, researchers have extensively analyzed the modular structure of the ITN. He et al. investigated the impact of globalization and recessions on the world trade network's hierarchical structure [6], while Nobi et al. explored the structural changes in the trade flow network of various commodities from 1995 to 2013 using the hierarchical path [7].

Interestingly, different reports have presented varied perspectives on the nature of the trade network. Some assert a trend towards globalization [8], while others have observed regionalization within the network [9,10]. There are also studies that argue for both viewpoints [11]. Furthermore, several works have focused on module detection and the regional trade effect in the world trade context [11–17]. One study utilized multi-regional input–output data to analyze the substructure of the international trade network across 186 nations and 26 industry sectors. Through the examination of interdependent networks and complex network metrics, key elements of the global economy were identified [12]. Another article focused on the dynamics of modules in the rare earth materials trade by constructing both unweighted and weighted networks. The study highlighted Japan, China, the USA, and Germany as major nations in this trade [13]. In a different research endeavor, del Río-Chanona et al. generated a secondary global trade network by correlating the weighted degrees of countries in the primary trade network. This approach revealed hidden module properties and showcased interesting deviations in modular structures between the two networks [14]. Chen et al. introduced a set of global and local module functions for module detection in networks, employing a form of self-loop rescaling technique. The study concluded the advantage of local modules in detecting modular structures on large-size and heterogeneous networks [15].

Using intraregional livestock trade data, Valerio et al. constructed networks of animal movements between West African regional markets. The identification of key trade modules and market roles led to the observation of both national and cross-border trade modules through module analysis [16]. Wang et al. adopted the trade in value-added accounting method to understand the influence of regional preferential trade agreements (PTA) on international manufacturing trade. Their analysis covered overall features, modular structures, and influencing factors, revealing a significant clustering effect in the international manufacturing trade in value-added networks and PTA networks over time [17–19]. Furthermore, a feature ranking network analysis on global and local indices demonstrated an increase in module and clustering indicators during crises [20,21]. Moreover, a mutual information-based minimum spanning tree revealed the grouping of regional countries together in a financial network [22].

In the complex global food system, the international food trade has been crucial determinants of food security. Clement et al. employed a module detection approach along with a supervised learning in the agricultural-food trade network [23]. Cong et al. analyzed the international energy trade network and detected modules evolving over time [24]. Wang et al. examined the evolution pattern of African countries oil trade under the changing in the global oil market [25]. They observed forming district trade modules which exhibit relative independence and self-reinforcement, influenced by factors such as the timing of market entry, international economic trends, energy demand, and geopolitical risks. Li et al. observed three modules in the international pesticide trade networks [26]. They detected some measures to decrease the properties of network over time. Cho et al. proposed a multiresolution framework that integrates information from a range of resolutions to consider trade communities [27]. The global trade networks were analyzed using exponential random graph models [28]. The multiple factors affected the global trade network. Kim and Yun compared a direct trade network to a trade network constructed using the personalized Page Rank [29]. Wang et al. investigated the potential risk shock propagation in global aluminum ore trade [30]. They observed significant change of trade community and the center of trade gravity. Chen et al. assessed the trade network connectivity of international trade networks along the belt and road [31]. They

observed the community structure, core–periphery structure, and backbone structure in the trade networks.

Examining the dynamic changes in the modular structure of different commodities can provide valuable insights into trade relationships among countries, thereby assisting policymakers and traders in formulating effective future trade strategies. The modular structure of a commodity trade network depicts the relationships between trading partners. These relationships are determined by the supply and demand for products between trading countries and are influenced by the global value chain. Trade networks can help you understand the relationships in these trade value chains and how they are affected by economic conditions.

In this article, we explore the evolution of modular structures in international trade by examining the trade matrices of 168 countries involved in trading eight products from 1995 to 2018. To achieve this, we construct trade networks and employ the Louvain algorithm to identify optimal modules within the trade network. Movements between modules in the trade network reflect the temporal variation in trade between each country. By doing so, we identify the most localized and globalized regions for trading all commodities and assess the strength of modules in the networks across different commodity markets. The analysis of module dynamics provides valuable insights into global trade trends, fostering sustainability in trade practices, and comprehending the impacts of crises on various commodities.

Additionally, we utilize the module core detection algorithm to identify key traders within modules in the commodity networks. Furthermore, we measure the influence of individual nodes on different commodity networks using an extended version of the module core detection algorithm. To gain a comprehensive understanding of the impact of financial events, we closely monitor and analyze market conditions under different financial crises. We observe and document the diverse effects of these crises on various commodity markets, and we identify sensitive products that are particularly affected during such periods.

## 2. Materials and Methods

We analyzed eight categories of products in world trade, classified as follows: (1) Food and live animals chiefly for food, (2) Beverages and tobacco, (3) Crude materials, inedible, except fuels, (4) Mineral fuels, lubricant, and related materials, (5) Animal and vegetable oils, fats, and waxes, (6) Chemical and related products, (7) Manufactured goods chiefly classified by material, and (8) Machinery and transport equipment. The annual trade volume data from 1995 to 2018 for 168 countries were collected from the United Nations Commodity Trade Statistics Database [18].

To create a comprehensive analysis, we constructed an $N \times N$ trade matrix for each dataset in each year, where $N$ represents the number of countries participating in the International Trade Network (ITN), which, in our case, is $N = 168$. The elements of these matrices represent the trade value in USD\$ between countries, and we aggregated all bilateral commodity flows between any two countries. To ensure accurate representation, the diagonal values in the matrices were set to zero since a country cannot export or import with itself.

Furthermore, to establish a normalized representation of the trade relationships, the elements of the trade matrix were divided by the sum of all trade values. This normalized matrix serves as the weighted adjacency matrix of the trade network, providing valuable insights into the trade relationships and interactions among the participating countries.

### 2.1. Module Optimization Algorithm

Modules, also referred to as clusters or communities, represent groups of nodes within a graph that likely share common properties or perform similar functions. Detecting these modules is crucial for identifying modules and their boundaries, enabling the classification of nodes based on their structural positions within these modules. The hub nodes of a

module play an important role in the network, and their importance is expressed in the network's connectivity. The roles of modules and hubs can be interpreted depending on the type of data that make up the network. On the other hand, nodes located at the boundaries between modules are crucial for facilitating mediation and maintaining relationships and exchanges between different modules [19].

To reveal the modular structure and assess the module strength, we utilize a quality function introduced by Newman and Girvan [32]. Module function enables us to quantify the extent to which the network's structure deviates from a random configuration, helping us identify and understand the presence of modules within the graph. The module function can be expressed as follows:

$$Q = \frac{1}{2m} \sum_{ij} (A_{ij} - P_{ij}) \delta(C_i, C_j), \tag{1}$$

where the sum runs over all pairs of nodes, $A$ is the adjacency matrix, and $A_{ij}$ represents the edge weight between $i$ and $j$. The value $m = \frac{\sum_{ij} A_{ij}}{2}$ corresponds to the total number of links in the graph, while $P_{ij} = \frac{s_i s_j}{2m}$ represents the expected number of links between nodes $i$ and $j$ in the null model. Here, $s_i = \sum_j A_{ij}$ denotes the strength of node $i$. The $\delta$-function yields a value of one if nodes $i$ and $j$ belong to the same module and zero otherwise. Consequently, the value of $Q$ lies between 0 and 1.

To determine the optimal partition of the network, we seek the modular structure that yields the maximum value of $Q$. In other words, a higher value of $Q$ indicates more significant deviations from the random counterpart [5]. The optimal partition represents the most suitable modular structure, providing valuable insights into the underlying patterns and relationships within the graph.

When analyzing both modular structures and identifying effective module cores, it becomes essential to optimize the module by determining the best partitions within the network. This optimization is necessary to detect optimal modules and observe shifts of nodes between predefined modules during different events.

To achieve this, we applied the Louvain algorithm, which operates in two phases, employing repeated iterations until we reach a global maximum of the module. The algorithm dynamically identifies stronger modules within the network. In the first phase, a partition of the network is created, with the number of modules initially equal to the number of nodes in the network. Subsequently, the algorithm iterates over all nodes and computes the module gain within the modules of their neighbors. If a positive variation in module is observed, the node movement is accepted. The change in module when inserting node $i$ into the module $C$ to which node $j$ belongs is represented as [33],

$$\Delta Q = \left[ \frac{\sum_{in} + 2k_{i,in}}{2m} - \left( \frac{\sum_{tot} + k_i}{2m} \right)^2 \right] - \left[ \frac{\sum_{in}}{2m} - \left( \frac{\sum_{tot}}{2m} \right)^2 - \left( \frac{k_i}{2m} \right)^2 \right], \tag{2}$$

where $\sum_{in}$ represents the sum of all the weights of links within the module $C$, $\sum_{tot}$ is the sum of all the weights of links incident to nodes in the module $C$, $k_i$ denotes the strength of the node $i$, which is the sum of weights coming to the node $i$, $k_{i,in}$ represents the sum of the weights of the links between $i$ and other nodes in the module $C$, and $m$ is the sum of the weights of all links in the network. When the change in module $\Delta Q$ is positive and increasing, we accept the move of node $i$ to a different module. However, if $\Delta Q$ does not increase, node $i$ remains in its original module. This process is repeated iteratively until a local maximum is reached.

In the second phase, known as the aggregation phase, all nodes in the same module are grouped together, forming a new network. In this new network, nodes represent the modules from the previous phase, and the weight of the new links between these nodes is determined by the total weight of the links between the original nodes that belong to the modules they come from [4]. When there are no more moves left that increase the

module, we reach a state with the global maximum of the module, signifying the optimized partition of the network into modules.

### 2.2. Module Core Detection Algorithm

We identify the optimal modules within a network based on the maximum module. However, it is important to note that nodes within each optimal module play different roles in the network. A node belonging to the "core" of a module contributes significantly to its stability. Therefore, removing such a core node can have a substantial impact on the network's partition. On the other hand, the removal of nodes that are closer to the module's boundary, i.e., nodes with fewer links inside the module, has a smaller effect compared to removing core nodes [4].

To detect the core of a module, we need to optimize the network partitions using the module. Nodes can shift across partitions due to field-specific events, which can only be detected in the network that has been optimally modularized. Consequently, it is necessary to recalculate the best modules before identifying the best core nodes. To do this, if we move a node from its module and consider placing it in other modules, we have $M - 1$ possible choices as targets, where $M$ is the number of modules. The node is placed in each targeted module alternately, and for each displacement, the module is recalculated. This process is performed for every node in a partition. We then determine the lowest absolute variation in a module achieved through the node's movement from its module. The node whose removal results in the most significant drop in a module is identified as the most influential node and thus the core of that partition. Each network partition will have its own core node [5].

### 2.3. Influence of a Node in the Network

The nodes within the network play distinct roles, and their influence is determined through an extension of the core detection algorithm. To identify the influential nodes, we select a node from a specific group and successively place it into other modules. The process begins by determining the primary group for a country based on the optimal modular structure. We then transfer the node from its primary module to other groups and calculate the resulting change in a module. The node that leads to the maximum change in a module is considered the most influential node. In other words, the node whose removal from its original group has the most significant impact on the network's module is identified as the most influential node.

The most influential node in the network is identified by following equation:

$$I_i = \frac{|dQ_i|}{\sum_{i=1}^{n}|dQ_i|} \times 100 \tag{3}$$

where $|dQ_i|$ represents the maximum change in a module when node $i$ is shifted to another module, and $\sum_{i=1}^{n}|dQ_i|$ denotes the total change in the whole network for $n$ nodes. This equation enables us to quantify the influence of each node and identify the node with the greatest impact on the overall network structure. The higher the $I_i$ of a node, the greater the node's influence in the network.

### 3. Results

Let us focus on understanding the evolution of the modular structure in the International Trade Network (ITN) based on individual products. Additionally, our aim is to identify the most influential countries in the ITN and examine the dynamics of their performance in trading individual products. To achieve this, we analyze the annual time series data of eight products in the international trade market for 168 countries over a 24-year period, spanning from 1995 to 2018. Subsequently, we categorize all countries into four modules based on their geographical locations in Asia-Oceania, Africa, America, and Europe.

Next, we employ the Louvain algorithm to identify potentially the best partitions that extract the optimal modular structure of the network. This allows us to track the displacement of nodes from their predefined modules and identify the core nodes within each optimized module. Finally, we measure the influence of these core countries on each commodity market to observe their roles during different financial events. By conducting this comprehensive analysis, we aim to gain valuable insights into the evolution of the modular structure in the ITN and the significant contributions of core countries to individual product markets over time.

### 3.1. Modular Structure Analysis

We quantified the module of the international trade networks each year using the Girvan–Newman algorithm. This algorithm is based on the trading relationships among countries, allowing us to observe how connections exist between them within the predefined modules. The algorithm evaluates the comparison between the actual density of realized connections and that of the predefined clusters.

The module value obtained from this analysis lies between 0 and 1. A high module value indicates that countries are predominantly engaging in intratrading within their respective modules. Conversely, a low module value suggests that countries from different modules exhibit strong interconnections within the networks. By measuring a module over time, we can gain insights into the extent of trade interactions within and between predefined modules, helping us understand the network's overall structure and dynamics.

Figure 1 illustrates two lines representing different module values in the International Trade Network (ITN). The blue (bottom) line shows the module achieved using the Newman–Girvan algorithm with predefined modules, while the green (top) line represents the optimized module obtained through the Louvain algorithm with optimized modules. The Newman–Girvan algorithm with predefined modules does not allow us to identify the best modular structure and optimal cores in the ITN. In contrast, the Louvain algorithm potentially provides the best network partitions, resulting in significantly a higher module compared to the Newman–Girvan algorithm with predefined modules. Consequently, the Louvain algorithm enables us to easily identify the best modular structure and cores in the ITN. Moreover, it helps us detect any shifts of countries from their predefined location-based modules, providing insights into the reasons for such displacements.

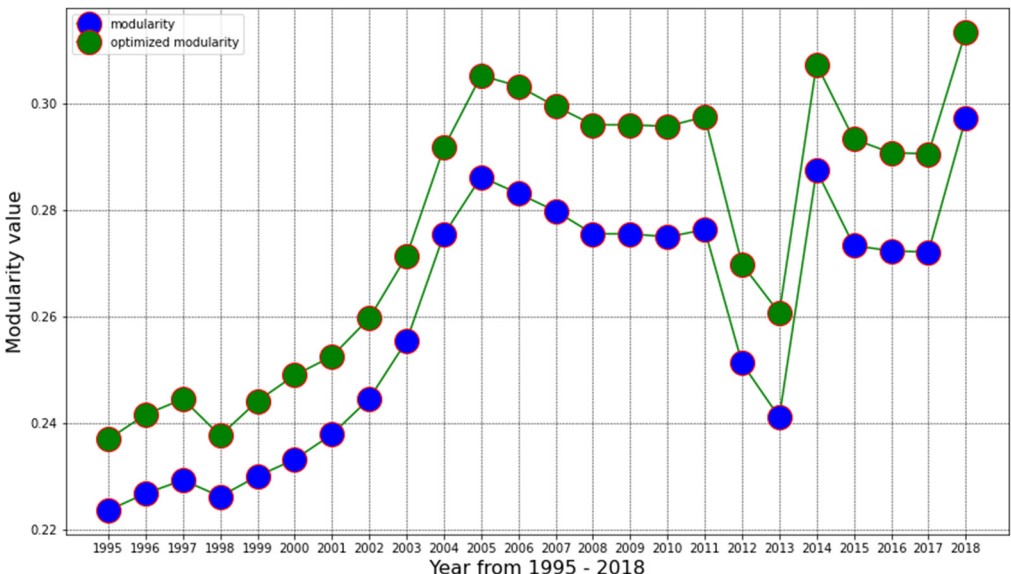

**Figure 1.** Comparison between the module values obtained from the predefined modules (blue curve) and the optimized modules (green curve) of the International Trade Network (ITN) across all commodities for the period 1995–2018.

Interestingly, we observe an increasing trend in the module of the ITN over the years, indicating a preference for regional trading. Another notable finding is the similar module trend for both predefined and optimized modules. The consistent change in module suggests that most countries tend to retain their predefined module positions in the network, and achieving the best partition mainly involves shifting a small number of nodes from their predefined modules. However, African countries stand out as they are frequently displaced from their predefined modules and merged with other modules, as shown in Table 1.

**Table 1.** The number of countries belonging to different modules for all commodities in selected years. The abbreviations AO, AF, AM, and EU represent Asia-Oceania, Africa, America, and Europe, respectively.

| Year | 1st Module | | | | 2nd Module | | | | 3rd Module | | | | 4th Module | | | |
|---|---|---|---|---|---|---|---|---|---|---|---|---|---|---|---|---|
| | AO | AF | AM | EU | AO | AF | AM | EU | AO | AF | AM | EU | AO | AF | AM | EU |
| 1998 | 14 | 26 | 3 | 37 | 13 | 10 | 0 | 0 | 29 | 1 | 29 | 0 | 0 | 0 | 6 | 0 |
| 2003 | 13 | 25 | 2 | 37 | 42 | 10 | 0 | 0 | 1 | 2 | 36 | 0 | | | | |
| 2013 | 11 | 18 | 1 | 37 | 44 | 18 | 0 | 0 | 1 | 1 | 37 | 0 | | | | |
| 2018 | 10 | 6 | 1 | 37 | 45 | 31 | 2 | 0 | 1 | 0 | 35 | 0 | | | | |

Overall, the pattern of modules in the ITN can be classified into two periods. The first period, from 1995 to 2005, shows an increasing phase, while the second period exhibits a plateau trend, with interruptions observed during the periods from 2012 to 2014 and 2018.

The uptrend observed in the first phase can be attributed to countries within a trade module engaging in greater trade among themselves. This phenomenon often occurs as countries seek to protect their domestic economies by forming higher-level groupings during times of recession [6,34]. Additionally, the increasing participation of countries in regional trade agreements (RTAs) has contributed to a more modular structure in the network.

The second period, spanning from 2006 to 2018, exhibits a plateau in both module curves, with interruptions occurring during the time periods of 2012 to 2014 and 2018. In these interrupting periods, the optimized module experienced a sudden decrease, reaching a local minimum in 2013, but it subsequently recovered to the plateau level in 2015, following the sharpest rise in 2014.

The decline in economic growth of EU countries in 2012 and their continued sluggishness in 2013, due to the ongoing recession in the Euro area, led to reduced trade among EU countries. During these periods, EU countries primarily traded within their module, while countries from other continents traded across all modules. Consequently, regional trade declined, resulting in a decrease in modules. However, in the following year, a remarkable rise in modules occurred, indicating a rapid shift from globalization to regionalization among continents during the 38 percent fall in global commodity prices [35]. The highest trading rate was recorded within the Asia-Oceania continent countries, highlighting their significant role in facilitating regional trade [36].

During the period from 2015 to 2017, world trade experienced a steady recovery, particularly when China emerged as the USA's largest trade partner. However, in 2018, the situation changed dramatically as the US began imposing tariffs on three "lists" of goods from China, triggering a trade war between the two economic giants. As a result, China retaliated with its own tariffs, leading to a significant decline in both US exports and imports to China. The impact of the trade war extended beyond specific industries, even affecting car sales in China, which, as the world's largest vehicle market, witnessed its first annual fall in twenty years during 2018. The China Association of Automobile Manufacturers (CAAM) attributed this decline to the trade war with the US. Moreover, China published its largest trade surplus in 2018, which was influenced by the trade war.

During this trade war in 2018, we observed a shift in African countries towards regionalization with the Asia-Oceania module. Notably, the instability resulting from the trade war was distinct from previous crises, such as the Asian financial crisis in 1998, the global financial crisis in 2008, and the European recession from 2012 to 2013. In those crises, modules experienced a downturn, indicating a decrease in regionalization. However, the instability caused by the trade war in 2018 showed a sharp rise in modules, pointing to a different pattern of regionalization. This indicates the regionalization of China and the USA with their allies during the trade war, reflecting the significant impact of the conflict on trade patterns and modular structures within the International Trade Network.

The trade networks constructed using the personalized Page Rank compared to the direct trade network [29]. Cho et al. proposed a multiresolution framework that integrates information from a range of resolutions to consider trade communities [27]. These results differ from current works because they do not employ the optimized Louvian algorithm.

Figure 2 displays the evolution of optimal modules over two different years, encompassing 56 Asia-Oceania countries, 37 African countries, 38 American countries, and 37 European countries. The continents' nodes are color-coded to aid in comprehending the modular structures of the ITN (International Trade Network). The modular structures for the years 2013 and 2018 are highlighted, representing the sharpest drop and peak in modules, respectively.

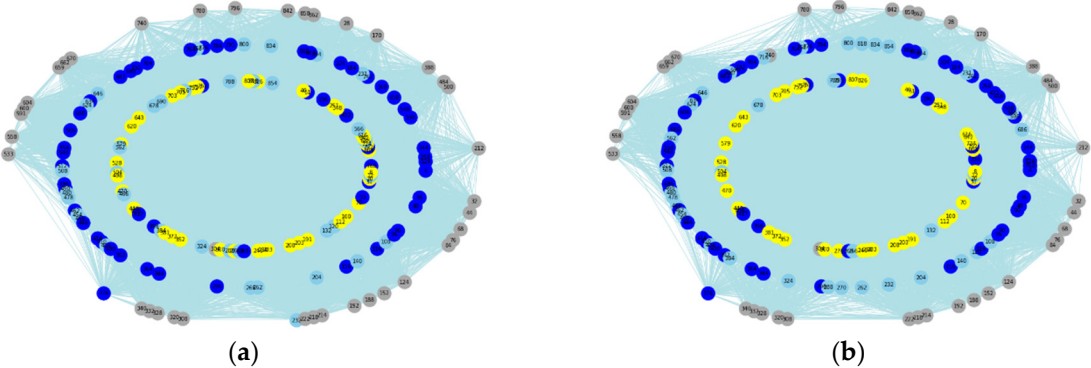

|    (a)    |    (b)    |

**Figure 2.** Modular structure of the International Trade Network (ITN) for all commodities. Each circle represents an individual module, with different continents shown in distinct colors: Africa in sky-blue, Asia-Oceania in blue, Europe in yellow, and America in dark gray. Specifically, (**a**) displays the modular structure in 2013, while (**b**) presents the modular structure in 2018.

In Figure 2a, the modular structure of 2013 is illustrated. During this year, only the countries of Europe remained in the same module, while countries from other continents were distributed across all modules, indicating a globalized state at that time. On the other hand, Figure 2b showcases the modular structure for the year 2018. The observed high module points to significant regionalization. This is attributed to the presence of three robust clusters: the European cluster, the American cluster, and a combined cluster of Asia-Oceania and African modules. Notably, the nodes from these three clusters experienced the least displacement during this year.

As can be seen in Figure 2 and Table 1, the trade of African countries forms two main modules. In Table 1, the first community consists of countries from Asia-Oceania, Africa, and Europe, while the second community is mostly composed of countries from Asia-Oceania and Africa. Many African countries have shown a tendency to transition from community 1 to community 2 over time. Due to the fragile international trade conditions of African economies, there is a tendency to rely more on regional trade, which is reflected in the concentration of trade within the community. A significant shift occurred in 2018 with the historic launch of the African Continental Free Trade Area (AfCFTA) at the African Union (AU) Summit, leading to increased regionalization as African countries aligned with the Asia-Oceania cluster. This shift played a crucial role in achieving the peak module in

2018. Table 1 presents the member counts of each module for selected years, illustrating a module downturn in 1998 and 2013, and an uptrend in 2003 and 2018.

For instance, in 1998, four modules were identified, with the first module comprising 14, 26, 3, and 37 members from the AO, AF, AM, and EU continents, respectively. This suggests that the majority of countries in the first module are from Africa and Europe, while the third module primarily consists of countries from America and Asia-Oceania.

The highest module for all commodities was observed in 2018. Table 1 reveals that most members in the first module are from Europe, while the third module consists predominantly of American countries. The second module shows the regionalization of African countries combined with the Asia-Oceania module, indicating increased trade among regional countries, and justifying the higher module during this year.

Throughout these four years, European countries consistently remained in their module, representing a regionalized pattern, while African countries displayed more dynamic interactions among.

We present the module scores for eight products in two subplots, as illustrated in Figure 3. Figure 3a displays the products with lower modules. First, let us focus on the modular structure for "Machinery and transport equipment", depicted by the yellow line. The module of this product shows slight fluctuations from 1995 to 2000, followed by a rapid increase from 2001 to 2004, with sharp local minima in 2013 and 2017. Initially, this product's trade was not confined to smaller regions, implying a lower module. However, from 2001 to 2004, it became more regionalized, maintaining relative stability in subsequent periods, except for two sharp falls from 2012 to 2013 and 2017. These declines can be attributed to the EU recession from 2012 to 2013 and the trade downturn from 2015 to 2016, which led to a more globalized trade in 2017.

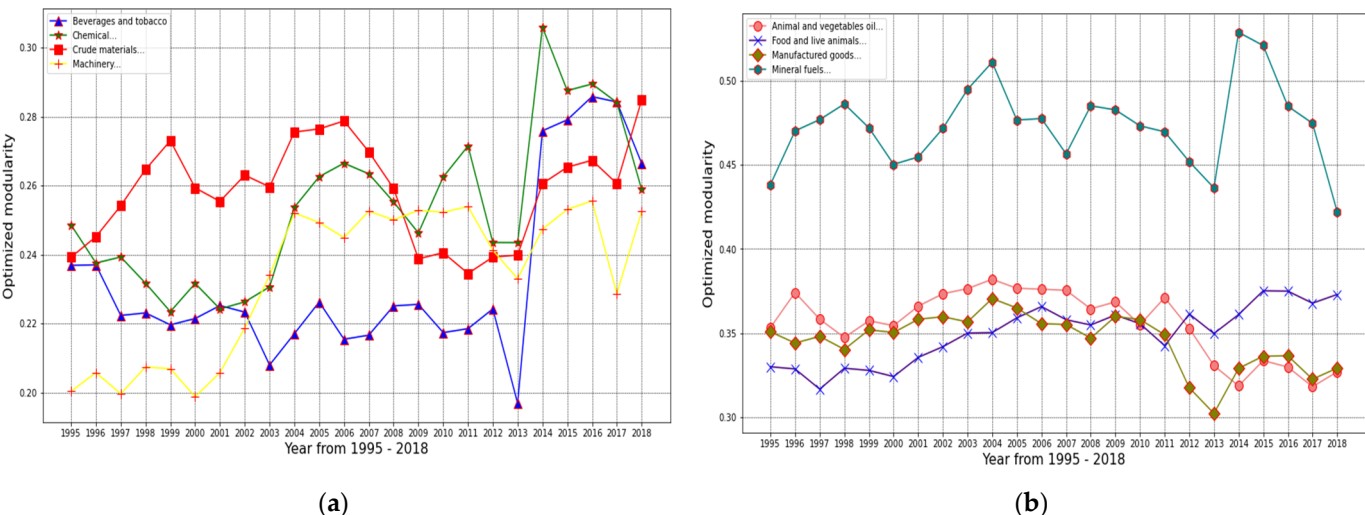

**Figure 3.** Modules for eight individual products from 1995 to 2018. The products were categorized into two groups: (**a**) Products (Beverages and tobacco (▲), Chemical and related products (★), Crude materials, inedible, except fuels (■), Machinery and transport equipment (+)) with low modules and (**b**) Products (Animal and vegetable oils, fats, and waxes (●), Food and live animals chiefly for food (×), Manufactured goods chiefly classified by material (♦), Mineral fuels, lubricant, and related materials (●)) with high modules.

Next, let us examine the modular structure of "Beverages and tobacco", represented by the blue line in Figure 3a. The module of this product remains relatively steady with minor fluctuations from 1997 to 2012. It experienced a drop in modules in 2013 due to the EU recession but witnessed a sharp rise in 2014, indicating a preference for trading in smaller regions. The structure continued to increase, except for a sharp fall in 2018, likely influenced by the US–China trade war.

Moving on to the green line in Figure 3a, we observe the evolution of the modular structure for "Chemical and related products". The module of this product shows a decreasing trend with minor fluctuations from 1995 to 2002, followed by a wave-like smooth shape between 2003 and 2009. Notably, it exhibits a local minimum in 2009 during the global financial crisis and again from 2012 to 2013 due to the EU recession, but it experienced a remarkable increase in 2014. The red line in Figure 3a represents the modular structure of "Crude materials, inedible, except fuels". This product's module follows an increasing trend from 1995 to 1999 and then fluctuates in a wave-like pattern until 2018, reaching a peak with a sharp rise during the USA-China trade war. Sharp decreases in modules are evident around global financial crises.

It is worth noting that all products in Figure 3a experienced a sharp uptrend in 2014, coinciding with a 38 percent fall in global commodity prices [35], which occurred just after the drop in the previous year because of the EU recession.

In Figure 3b, we present the products with higher modules, except for mineral fuels. For these products, we observe relatively small fluctuations in module. The dynamics of "Animal and vegetable oils, fats and waxes" and "Manufactured goods chiefly classified by material" products initially increased with minor fluctuations until 2004. Subsequently, they experienced a slow decrease with a sharp fall from 2012 to 2013 due to the EU recession. After this period, modules increased again and remained relatively steady during subsequent periods.

The modular structure for "Food and live animals chiefly for food", represented by the blue line, demonstrates a consistent increase over time with minor fluctuations. This suggests that the trade of this product is becoming more regionalized, which is reasonable as these food items are essential for daily life and would become expensive if traded over long distances. Notably, no sharp changes in modules are observed during different crises.

Let us consider the modular structure of "Mineral fuels, lubricant, and related materials", depicted by the teal line. We notice that after reaching local maxima in modules in 1998, 2004, 2008, and 2014, the module gradually decreased in the subsequent years before rising again. This product exhibits higher modules than any other product, with its peak module observed after the EU recession in 2014, during the commodity price shock [35]. The trade of this product appears to be more regionally dependent compared to other products.

We reveal that "Mineral fuels, lubricant, and related materials" achieved the highest module value, while "Machinery and transport equipment" attained the lowest. The significant cost and complexity involved in setting up and maintaining fuel supply lines over long distances led to the regional constraint of this product's trade, resulting in its highest module.

In contrast, "Machinery and transport equipment" is a product that many countries prefer to import rather than produce and export due to production complexity and risk factors. This creates a global market for the product, hence its lower module. Throughout different financial crises, especially the EU recession, most products experienced globalized trade markets. However, the sharp rise in modules during the commodity price shock in 2014 and the trade war in 2018 indicates a shift towards a regional market structure, implying that these two crises had different effects on the market dynamics. Additionally, a notable decline in modules was observed for chemicals, beverages, and fuels in 2018, indicating varying impacts of the trade war on these specific products.

### 3.2. Cores of the Modules

During our observed period, we identified three to four optimized modules, as presented in Table 2. Generally, there were three modules, except for the years 1998, 2005, 2006, and 2011, where four modules were observed.

**Table 2.** Optimal core modules using by core detection algorithm for each year. The USA consistently remains at the core of a module. However, the status of Germany, Japan, and China has changed.

| Year | Module Cores | Year | Module Cores |
|------|--------------|------|--------------|
| 1995 | USA, Germany, Japan | 2007 | USA, Germany, China |
| 1996 | USA, Germany, Japan | 2008 | USA, Germany, China |
| 1997 | USA, Germany, Japan | 2009 | USA, Germany, China |
| 1998 | USA, Germany, Brazil, India | 2010 | USA, Germany, China |
| 1999 | USA, Germany, Japan | 2011 | USA, Germany, China, Togo |
| 2000 | USA, Germany, Japan | 2012 | USA, China, Italy |
| 2001 | USA, Germany, Japan | 2013 | USA, China, Italy |
| 2002 | USA, Germany, China | 2014 | USA, Germany, China |
| 2003 | USA, Germany, China | 2015 | USA, Germany, China |
| 2004 | USA, Germany, China | 2016 | USA, Germany, China |
| 2005 | USA, Germany, China, Zambia | 2017 | USA, Germany, China |
| 2006 | USA, Germany, China, Zambia | 2018 | USA, Germany, Japan |

The USA consistently belongs to the core of one module, as indicated in Table 2. The core position refers to the node with the most connection in each module and corresponds to the local hub in the module. However, the status of Germany, Japan, and China has undergone changes over the years. Germany consistently belonged to the core of the European module, except in 2012 and 2013, during the time of the EU recession when Italy took its place. Japan held the core status from 1995 to 2001, after which it shifted to China in 2002. This transition was influenced by the so-called Lost Decades in Japan, lasting from 1991 to 2011 [37]. However, following the implementation of Abenomics under Prime Minister Abe's leadership in 2012, which involved policies like negative interest rates, unlimited currency supply, and 2–3% inflation, Japan regained its core position in the year 2018.

Occasionally, Brazil, India, and Italy appeared in the core of some modules. Moreover, Zambia and Togo occupied the core position of the African module for certain years. From 1995 to 2001, Japan held the core position within the Asia-Oceania module, except for the year 1998 when this module merged with the American module under the governance of the USA, owing to the Asian financial crisis from 1997 to 1998. During that period, India became the core of a smaller module comprising 23 members, with 13 members from Asia-Oceania and 10 from Africa. Additionally, Brazil governed another small module with only 6 American members that year. The local hub changes dynamically according to changes in trade. This change means a change in trade relations, and a change in the local hub means a large change in trade relations. Countries connected to a local hup form a strong trade correlation.

The Asian financial crisis quickly recovered from 1998 to 1999, and since then, Asia-Oceania has been an independent module ruled by either China or Japan. From 2005 to 2006 and 2011, small modules consisting of only three to four members from the African continent emerged, with Zambia and Togo taking charge in different periods for those specific years. However, over time, these modules merged with other larger modules.

Throughout the years from 1995 to 2018, the Europe module was consistently governed by Germany, with one exception from 2012 to 2013. During this period, Italy experienced high GDP growth and achieved its most significant export and import values, leading to Italy assuming the core position and replacing Germany. Nevertheless, this change was short-lived as Germany regained its core position from 2014 onwards and continued to rule the Europe module.

Modules were extracted from a network where trade connections exist between two countries, rather than being based on trade volume. In other words, the network used to extract the modular structure is not a weighted network. Therefore, in each module, the core country may not necessarily be the country with the highest total trade volume. The node with the core position is at the center of connectivity within the module. The node in the core position is similar to the local degree centrality node within the local module since it has the highest degree. While degree centrality nodes are defined in the entire network, core position nodes are defined within the module. Therefore, countries within the module have shorter paths connected to core position nodes, reflecting the efficient flow of trade between core position nodes and other nodes within the module in the network.

Most of the core positions measured were the USA, Germany, China, and Japan, which are central countries in world trade. Therefore, most countries connected to core position nodes have active trade relationships with each other. After checking the modules and core positions, we looked at the trade volume between the two countries and found that the trade volume between the two countries was large. Of course, countries with small trade volume, such as African countries, also have a strong tendency to be connected to the core position.

Consequently, countries other than the United States, Germany, Japan, and China may temporarily emerge as core modules. For instance, Togo and Zambia may emerge as core countries when the fourth module appears, and the countries within the fourth module are relatively few, consisting of African and American nations. Therefore, Togo and Zambia could be core countries of the fourth module.

Figure 4 illustrates the influence of cores on individual products in world trade, measured by Equation (3). The dominant cores in this analysis are the USA, Germany, China, and Japan, as they frequently occupy core positions. In Figure 4a, we observe that for "Food and live animals chiefly for food", the USA exerts the strongest influence in the network compared to the other three cores. Throughout the years, we notice very similar change patterns in the influence of these cores.

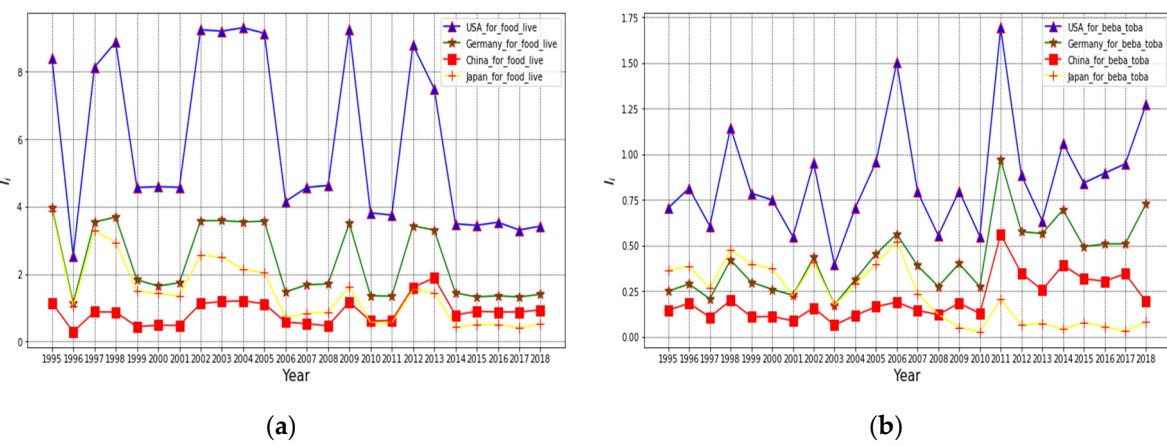

(**a**)          (**b**)

**Figure 4.** *Cont.*

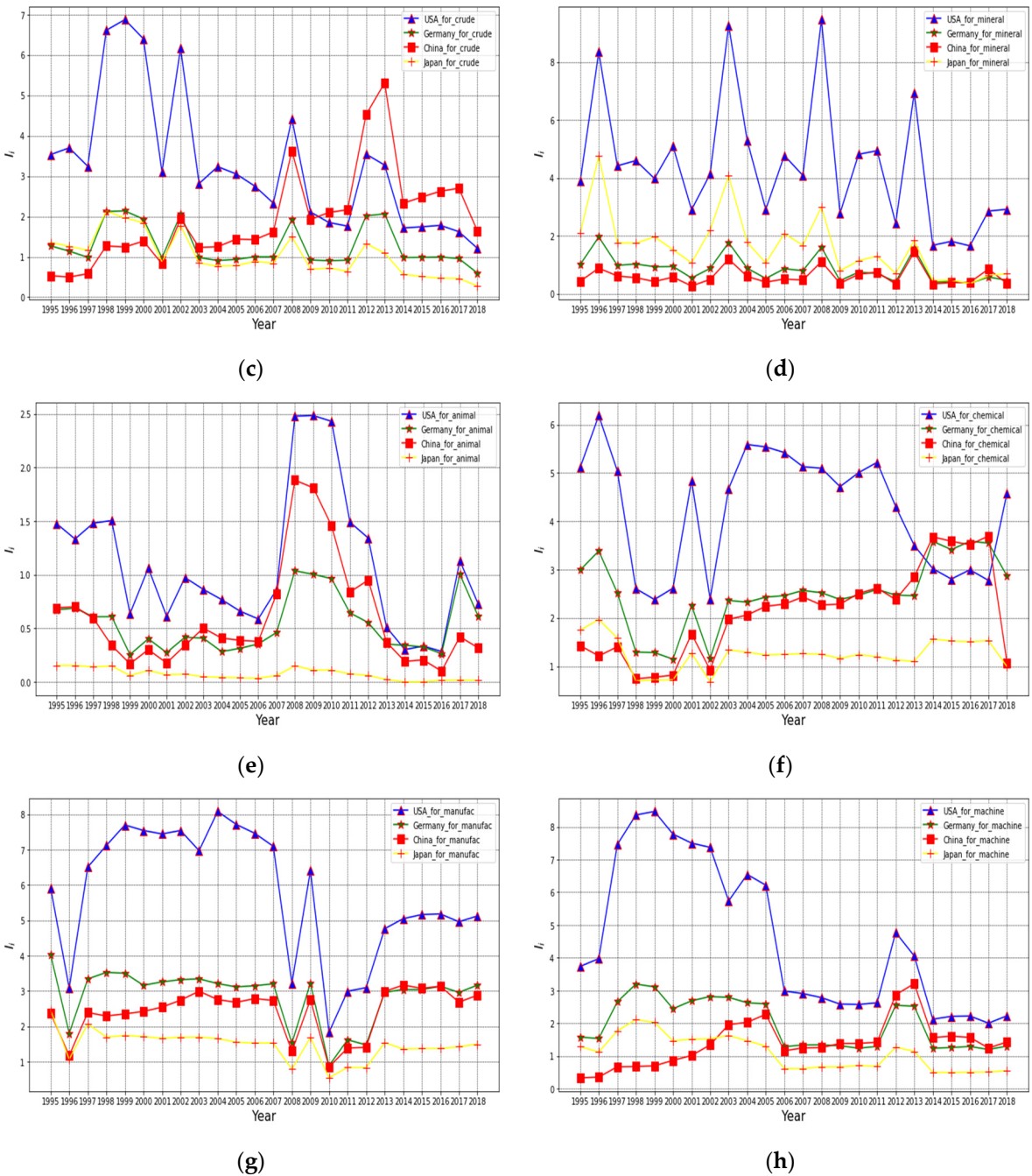

**Figure 4.** Influence of the cores on individual products. (**a**) Food and live animals chiefly for food, (**b**) Beverages and tobacco, (**c**) Crude materials, inedible, except fuels, (**d**) Mineral fuels, lubricant, and related materials, (**e**) Animal and vegetables oil, fats, and waxes, (**f**) Chemical and related products, (**g**) Manufactured goods chiefly classified by material, (**h**) Machinery and transport equipment.

The most significant rises in the influential position of these cores occurred in 1997, 2002, 2009, and 2012 for this product, possibly due to the rise in food insecurity in Asia during those periods. Conversely, in 1996, 1999, 2006, 2010, and 2014, all the cores experienced a noticeable downfall in their influence, with the USA, although still the highest influencer in the world trade network for this product during those years, facing a considerable decline. From 2014 to 2018, a flat line is observed for all cores, indicating no significant change in their market position during the trade war for this product.

In Figure 4b, we observe the influence of cores on "Beverages and tobacco", where the USA remains the most influential core throughout the years, despite experiencing many

ups and downs in its journey. Japan initially held the second most influential position from 1995 to 2003 but gradually declined and became the least influential core from 2009. The highest rise in influence for all cores is evident in 2011.

For "Crude materials, inedible, except fuels" (Figure 4c), the USA initially held the most influential core position, but it gradually declined with some significant fluctuations around different crises. China, on the other hand, started from the lowest position but steadily improved its status, becoming the top influential core in trading crude in 2010 and retaining its leading position thereafter. All core countries became less influential during the trade war.

Figure 4d depicts the cores of the trade network for "Mineral fuels, lubricant, and related materials". Similar to other products, the USA was the most influential core in trading minerals throughout the monitored period. The influential patterns for all cores show similar ups and downs. Japan initially held the second most influential position, but after 2013, it coincided with Germany and China.

In Figure 4e for "Animal and vegetable oils, fats, and waxes", all module cores, particularly the USA and China, significantly increased their influence during the 2008–2009 global financial crisis. The second most influential position alternated between China and Germany, but after 2013, Germany surpassed the USA and remained as the top influential trader. Japan's influence remains almost flat at the bottom of the plot.

In Figure 4f, for "Chemical and related products", we observe that China started at the bottom but gradually rose alongside Germany over the years, eventually reaching the top position in 2014 and maintaining it until 2017. Before that, the USA held the greatest influential core position, with significant dominance from 2003 to 2011. However, it experienced a downturn and lost its top position to China and Germany in 2014, while the other three cores also rose and remained at that level from 2014 to 2017. During the trade war in 2018, these three cores, especially China, encountered a sharp fall, while the USA made a jump and reclaimed the top position.

Figure 4g illustrates that the USA is the most influential core in the network of "Manufactured goods chiefly classified by material", displaying exclusive dominance between 1997 and 2007. The trend of influence is somewhat similar for all cores. Notably, every core experienced sharp falls in 1996, 2008, and 2010, and rose sharply in the following years during the Asian financial crisis, the global financial crisis, and the EU recession, respectively.

Upon examining the influence of cores on various commodities, we observe an almost similar trend pattern, indicating strong communication among cores of different modules, particularly during crises. Throughout the trade of most products, the USA stands as the dominant force, China emerges significantly in this century, Germany maintains stability, while Japan experiences a decline in influence.

Interestingly, cores tend to exhibit identical responses during crises, showing cohesive behavior. However, during the trade war, different reactions are observed for some products. We applied network analysis that crisis events had a significant impact on the changes in ITN modules. While it can be speculated that economic crises may have had a substantial impact on global trade, the use of complex network analysis was employed to investigate the changes in ITN modules.

## 4. Discussion

Countries interact strongly with each other through trade, with trade flows influenced not only by their economic situations but also by their geographical locations. To understand these dynamics, we categorize countries into four groups based on continental distribution and observe their transformation after optimal modularization.

Remarkably, we find a similar module trend for both predefined and optimized modules, indicating that most countries tend to retain their regional module positions in the network, with the best partition achieved by only shifting a small number of nodes from their predefined modules. However, African countries show a distinct pattern, being most often displaced from predefined modules, and merged with other modules.

Analyzing the module curves of all commodities, we observe two patterns. The first phase shows a rising trend of regional trade from 1995 to 2005, followed by a plateau pattern with interruptions in 2012 to 2014 due to the EU recession and in 2018 due to the trade war. During all crises, including the Asian financial crisis from 1997 to 1998, the global financial crisis from 2008 to 2009, and the European recession from 2012 to 2013, the module shows a downturn. However, a sharp rise is noticed during the trade war in 2018, driven by strong groupings of nations with China and the USA. By observing module formation, we consistently find three optimal modules in most years, with European countries being the most stable in their regional module, while African countries exhibit a more globalized nature.

The modular structure extracted from the world trade network was observed to change depending on the economic environment. Changes in modularity could be observed before and after the economic crisis. Meanwhile, the modularity fluctuations of "Animal and vegetable oils, fats and waxes", "Manufactured goods chiefly classified by material", and "Food and live animals chiefly for food" fluctuate relatively small over the entire period. These items maintain stable trade relations. On the other hand, items excluding these were highly volatile. For highly volatile items, changes in trade relations intensified due to the economic environment and economic crisis. Linking these changes to trade stability and sustainability is left for further work.

Furthermore, we observe the trend of modules for different products over time and note that the "Mineral fuels, lubricant, and related materials" product displays the highest modular structure, while "Machinery and transport equipment" exhibits the lowest. The module of different products experiences sharp transitions during economic recessions. Human-made products, such as Machinery, Beverages, and Manufactured goods, are found to be more responsive to crisis shocks compared to natural products. We also demonstrate the changes in the modular structure of different products due to the US-China trade war.

Using the module core detection algorithm, we consistently identify core countries in optimal modules each year, with the USA, Germany, China, and Japan often occupying core positions. When observing the influence of cores on various commodities, we notice a nearly identical trend pattern for each product. This indicates effective communication among cores of different modules, especially during crises. In the trade of the majority of products, we observe the USA's supremacy, the emergence of China in this century, the stability of Germany, and the decline of Japan.

Interestingly, cores exhibit an identical response during crises, showing cohesive behavior. However, during the trade war, different reactions are observed for some products. The dynamics of modules in commodities networks can be useful in understanding the trend of exports and imports of different continents worldwide and in observing the effect of crisis shocks on the trade network modules for various commodities. We can further extend our analysis on the other time series such stock indices, foreign exchange market, and futures. The ITN has been shaped by various factors over the long period. However, since ITN is a nearly fully connected network which extracted from the trading connectivity among countries, we ignored factors such as total trade volume and the weighted effects based on trade between two countries, which has led to limitations in our research. We only considered the change of the internal network structure of ITN by the module detection using the Louvain module optimization method.

This paper does not delve into the relationship between the modularity changes discovered in this study and the significant changes in the global value chain. This aspect could be a subject of future research for the authors. The sensitivity test for trade volume, is indeed an intriguing process, we leave it as a potential area for future research as well. The core countries extracted in this paper may not necessarily play a major role in trade. However, observing changes when breaking down the international trade network using complex network methods and relating these changes to other elements of international trade is a topic for future research.

**Author Contributions:** Conceptualization, Z.M.K. and A.N.; methodology, Z.M.K., M.I.R. and J.A.; software, Z.M.K., M.I.R. and J.A.; validation, Z.M.K., M.I.R. and J.A.; formal analysis, Z.M.K. and A.N.; investigation, Z.M.K., M.I.R. and J.A.; resources, A.N. and J.W.L.; data curation, Z.M.K. and A.N.; writing—original draft preparation, Z.M.K. and A.N.; writing—review and editing, Z.M.K., J.W.L. and A.N.; visualization Z.M.K., M.I.R. and J.A.; supervision, A.N. and J.W.L.; project administration, A.N.; funding acquisition, A.N. and J.W.L. All authors have read and agreed to the published version of the manuscript.

**Funding:** This work was supported by the Ministry of Education of the Republic of Korea and the National Research Foundation of Korea (NRF-2020R1A2C1005334). This research is also supported by university grant commission of Bangladesh and the ministry of ICT of Bangladesh (20FS34668).

**Institutional Review Board Statement:** Not applicable.

**Informed Consent Statement:** Not applicable.

**Data Availability Statement:** Publicly available datasets were analyzed in this study. This data can be found in UN Comtrade Database: [https://comtradeplus.un.org/, accessed on 6 January 2023].

**Conflicts of Interest:** The authors declare no conflict of interest.

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
