# Peer review of "Modular Structures of Trade Flow Networks in International Commodities"

_sustainability, doi:10.3390/su152215786_

Round 1
Reviewer 1 Report
Comments and Suggestions for Authors
The English language needs minor corrections.
Author Response
Reply to first reviewer
Reply: Thank you for your good comments.
It is my pleasure to review the current manuscript having title “Modular Structures of Trade Flow Networks in International Commodities” for the esteemed journal. The manuscript needs to be improved significantly to make sure the quality. The authors must consider the following suggestions:
- The recommendation of the study should be mentioned in the abstract based on the results of the study.
Reply: We modify the abstract reflecting the results.
- The novelty of this paper should be further justified by comparing the current paper with the existing ones. The literature review is not thick enough, Therefore, recent and relevant papers should be cited till 2023.
Reply: We include some current papers and compared them in line 85-104.
- In the methodology section, the model of the study needs to be justified by comparing it with other models. Moreover, a flow chart of the methods used in the study should be given.
Reply: The Louvian algorithm was explained in detail in reference 24. We used the algorithm of this reference. We only cited this reference and explained the algorithm in section 2. (24. Blondel, V. D.; Guillaume, J. L.; Lambiotte, R.; Lefebvre, E. Fast unfolding of communities in large networks. J. Stat. Mech. Theory Exp. 2008, P10008).
- In the results section, the obtained results should be compared with existing studies in the field.
Reply: We discuss some related works in this field. We include new references [27-31]. We add a paragraph in the results section in line [348-351].
Grammar check is required to avoid any possible English errors.
Reply: We checked English grammar again.

Reviewer 2 Report
Comments and Suggestions for Authors
Comment on “Modular Structures of Trade Flow Networks in International Commodities”
This is as far as I can see a competent application of the ITN model on aggregate trade and eight broad product groups. I am not qualified to comment in detail on the specifics of the presentation of the model or the modelling exercise itself (since I have not developed any ITN models from scratch), but I think it suffers from some of the same weaknesses that I have noted in earlier contributions using the same model. In general, these include a lack of explicit attention to trade theory, questionable interpretations of the empirical data, and a very short time perspective that seems to contradict the idea that a trade network is almost per definition a long-term construct based on substantial investment in market-specific knowledge and intra-firm relationships.
Beginning with the first of these issues, I think it would be useful to acknowledge that trade is not undertaken by states that seek to “rule” or “govern” others in bilateral relations or communities, but rather by companies that are driven by profit motives (with the possible exception of state trading nations such as China, where it is sometimes unclear whether decisions are driven by short-term profit motives or longer-term political objectives). Given the nature of trade costs (which are largely determined by distance, be it geographic, institutional, political, linguistic, historical, or psychic), it is natural that most trade takes place within regions, in particular if regional markets are large enough to absorb exports. The gravity model of international trade (commonly used for empirical analysis) neatly sums up these major drivers of trade flows. It is therefore surprising to see that authors use terms as “interesting” (line 269) and “remarkable” (line 545) to find high modularity both for the predefined and optimized regions. Many trade theories would give this prediction, and it would be appropriate to note that.
Furthermore, even the simple gravity model could contribute significantly to the observations concerning African economies. In many ways, the “distances” and trade costs are larger and the gravity pull is weaker in trade between African economies than in trade between individual African economies and their bilateral trade partners in the EU, the US, China, or the Middle East. It is misleading to argue that African economies show “more dynamic interactions among modules” and that they should therefore be considered “more globalized” (line 363-364), or that there is a “consistent trend of African countries exhibiting higher division and global trade” (line 343-344). African economies are more dependent on inter-regional trade because the conditions for intra-regional trade are so weak.
The empirical interpretations of the model results are sometimes a bit exaggerated because of the lack of attention to issues such as foreign direct investment. In many industries, perhaps in particular beverages and tobacco (but also sectors such as transportation equipment), foreign markets are primarily served through FDI. Trade can be a complement at times, depending on exchange rates and demand and supply conditions in both the source and destination economies. The trade data may therefore suggest that markets are highly volatile, while in reality consumption and production volumes may be fairly stable. Moreover, the attribution of all changes in data to a small number of events (the Asian Financial Crisis, the EU recession, the Global Financial Crisis, and the US-China Trade War) is not convincing. Although each of these have some effects in different regions (in particular, the impacts of the three first ones where by and large contained to the regions where they emerged) there are many other developments in effect. For example, macroeconomic imbalances and trade deficits in many of China’s Western export destinations would have reduced Chinese export growth (and hence raised modularity) even in the absence of a trade war between the US and China. Few European countries would have been able to increase imports from China given the bilateral trade deficits.
My biggest concern is perhaps the large fluctuations in the estimated values presented in the tables. Figure 1 is to me the most convincing, with relatively clear trends and a disturbance between 2011 and 2015 (where we see reversal to the trend from 2015). However, the fluctuations in the more disaggregated data (commodities and core countries) simply do not make any sense to me. Foreign market entry is costly and firms that have successfully established a foreign market position will typically try to maintain it for long periods. Much of world trade – probably more than one-third –is intra-firm trade within multination enterprises; it possible that trade in global value chains adds another third to the total. Although things change over time, the underlying fluctuations in market shares and production are rarely as large as those implied by Figures 3 and 4. My guess is that a large share of the fluctuations are due to changes in exchange rates and prices rather than trade volumes: it would be interesting to see some sensitivity testing with trade volumes rather than trade values. (See lines 297-298 on the 38% drop in commodity prices when modularity increased in 2014.)
The changes in the core communities are also puzzling. I understand the broad patterns, but it is not intuitively clear how Italy could become a core European economy in 2012-13, although its total trade volumes never reached half of Germany’s, or how Japan could become a community core in 2018, although China’s exports were 3-4 times larger. Moreover, the observations related to Zambia and Togo do not match any of my empirical observation regarding their role in African trade. Their total trade volumes are small in comparison with larger economies such as South Africa and Egypt, and even though the share of exports going to other “community” members may at times be higher, they can hardly be said to have a “leading role”. More arguments would be needed to make the analysis convincing,
Some smaller comments follow:
I was occasionally confused regarding the terminology, For example, it is not clear to me whether the terms “community” and “module” are synonymous. If so, it would be better to stick to one of them. The term “region” was used to refer to the years 1995-2005 and 2006-2018, respectively. Why not “time period”?
Line 81-82: It is stated that “Identifying the cores of communities and assessing their influence on various products can aid countries in selecting efficient and profitable trade partners.” How? Trade is typically determined by the competitiveness of firms – why would they be more interested in some partners than others? How are the active agents in international trade – the exporting companies – helped by this?
Line 87-88: It is stated that the analysis helps to understand “how countries shift from their regional communities due to the optimization of the modularity function based on the Newman-Girvan algorithm”. What does this mean? Were the countries aware of these shifts, or did they occur inadvertently?
Line 124-125: The text argues that “Nodes with a central position in their clusters, meaning they have numerous links with other group partners, are likely to play essential roles in control and stability within the group”. It is not clear to me how this control would be exercised.
I am bit confused by the meaning of “core position” in the community. Terms like “rule”, “govern”, “dominate”, “influence”, and “take charge” are used throughout the paper. What does this mean? How does a core country “rule”? Is it through its influence in regional trade agreements, or more covertly, through grand strategies and unilateral actions to maximize its soft or hard power?
Are the vertical axes in Figures 3 and 4 defined the same way? It is confusing that the scales are so different, centered around 0.3 in Figure 3 and up to 50 times larger in Figure 4. An explanation is warranted.
Line 535-536: What is the meaning of the sentence “The food, mineral fuels, manufactured goods, and machinery equipment markets demonstrate a greater influence scale.”
Comments on the Quality of English LanguageOverall, with few exceptions, the English is very good.
Author Response
Reply to second reviewer
Reply: Thank you for your good comments.
Comment on “Modular Structures of Trade Flow Networks in International Commodities”
This is as far as I can see a competent application of the ITN model on aggregate trade and eight broad product groups. I am not qualified to comment in detail on the specifics of the presentation of the model or the modelling exercise itself (since I have not developed any ITN models from scratch), but I think it suffers from some of the same weaknesses that I have noted in earlier contributions using the same model. In general, these include a lack of explicit attention to trade theory, questionable interpretations of the empirical data, and a very short time perspective that seems to contradict the idea that a trade network is almost per definition a long-term construct based on substantial investment in market-specific knowledge and intra-firm relationships.
Reply:
I agree to your comments. When applying the concept of complex systems to ITN, we agree with the weaknesses pointed out by the reviewer. It is also a valid claim that ITN has formed over the long period. The reviewer's comments, as pointed out, are added to the discussion section in one paragraph. However, this paper aims to understand the internal network structure of ITN, which has been shaped by various factors over the long period. Since ITN is a nearly fully connected network which extracted from the trading connectivity among countries, we ignored factors such as total trade volume and the weighted effects based on trade between two countries, which has led to limitations in our research. We added some sentences in the lien 611-618.
Furthermore, even the simple gravity model could contribute significantly to the observations concerning African economies. In many ways, the “distances” and trade costs are larger and the gravity pull is weaker in trade between African economies than in trade between individual African economies and their bilateral trade partners in the EU, the US, China, or the Middle East. It is misleading to argue that African economies show “more dynamic interactions among modules” and that they should therefore be considered “more globalized” (line 363-364), or that there is a “consistent trend of African countries exhibiting higher division and global trade” (line 343-344). African economies are more dependent on inter-regional trade because the conditions for intra-regional trade are so weak.
Reply:
We agree the reviewer’s opinion. As seen in Figure 2 and Table 2, African countries are concentrated in two modules. As Table 2 indicates, Community 1 connects countries from Asia-Oceania, Africa, and Europe, while Community 2 includes a substantial presence of Asia-Oceania and Africa. When we transition from 1998 to 2018, there is a tendency for African countries to shift from Community 1 to Community 2. The movement between modules does not imply that African countries need to become more globalized, as suggested by the reviewer. Instead, the trade among African countries in two modules are more dependent on inter-regional trade. We have addressed the reviewer's concerns.
Added sentences in results part in line 377-383 and line 399-401.
The empirical interpretations of the model results are sometimes a bit exaggerated because of the lack of attention to issues such as foreign direct investment. In many industries, perhaps in particular beverages and tobacco (but also sectors such as transportation equipment), foreign markets are primarily served through FDI. Trade can be a complement at times, depending on exchange rates and demand and supply conditions in both the source and destination economies. The trade data may therefore suggest that markets are highly volatile, while in reality consumption and production volumes may be fairly stable. Moreover, the attribution of all changes in data to a small number of events (the Asian Financial Crisis, the EU recession, the Global Financial Crisis, and the US-China Trade War) is not convincing. Although each of these have some effects in different regions (in particular, the impacts of the three first ones where by and large contained to the regions where they emerged) there are many other developments in effect. For example, macroeconomic imbalances and trade deficits in many of China’s Western export destinations would have reduced Chinese export growth (and hence raised modularity) even in the absence of a trade war between the US and China. Few European countries would have been able to increase imports from China given the bilateral trade deficits.
Reply:
Your comments are correct. This research is not intended to attribute all changes to the Asian financial crisis, EU economic downturn, global financial crisis, and US-China trade war. It is meant to illuminate through network analysis that these events had a significant impact on the changes in ITN modules. While it can be speculated that economic crises may have had a substantial impact on global trade, the use of complex network analysis was employed to investigate the changes in ITN modules. We added some sentences in line 573-577.
My biggest concern is perhaps the large fluctuations in the estimated values presented in the tables. Figure 1 is to me the most convincing, with relatively clear trends and a disturbance between 2011 and 2015 (where we see reversal to the trend from 2015). However, the fluctuations in the more disaggregated data (commodities and core countries) simply do not make any sense to me. Foreign market entry is costly and firms that have successfully established a foreign market position will typically try to maintain it for long periods. Much of world trade – probably more than one-third –is intra-firm trade within multination enterprises; it possible that trade in global value chains adds another third to the total. Although things change over time, the underlying fluctuations in market shares and production are rarely as large as those implied by Figures 3 and 4. My guess is that a large share of the fluctuations are due to changes in exchange rates and prices rather than trade volumes: it would be interesting to see some sensitivity testing with trade volumes rather than trade values. (See lines 297-298 on the 38% drop in commodity prices when modularity increased in 2014.)
Reply:
In Figure 2, there is a significant variation in ITN modularity between 2011 and 2015. We have not yet been able to ascertain the exact reasons for this variation. By examining the modularity changes of eight individual products in Figure 3, we attempted to identify the factors that had a significant impact on the overall modularity change. This paper does not delve into the relationship between the modularity changes discovered in this study and the significant changes in the global value chain. This aspect could be a subject of future research for the authors. The sensitivity test for trade volume, as suggested by the reviewer, is indeed an intriguing proposal, and we leave it as a potential area for future research as well. We added some sentence in line 624-630.
The changes in the core communities are also puzzling. I understand the broad patterns, but it is not intuitively clear how Italy could become a core European economy in 2012-13, although its total trade volumes never reached half of Germany’s, or how Japan could become a community core in 2018, although China’s exports were 3-4 times larger. Moreover, the observations related to Zambia and Togo do not match any of my empirical observation regarding their role in African trade. Their total trade volumes are small in comparison with larger economies such as South Africa and Egypt, and even though the share of exports going to other “community” members may at times be higher, they can hardly be said to have a “leading role”. More arguments would be needed to make the analysis convincing,
Reply:
In this paper, communities were extracted from a network where trade connections exist between two countries, rather than being based on trade volume. In other words, the network used to extract the community structure is not a weighted network. Therefore, in each community, the core country may not necessarily be the country with the highest total trade volume. Consequently, countries other than the United States, Germany, Japan, and China may temporarily emerge as core communities. For instance, Togo and Zambia may emerge as core countries when the fourth community appears, and the countries within the fourth community are relatively few, consisting of African and American nations. Therefore, Togo and Zambia could be core countries of the fourth community.
As the reviewer pointed out, the core countries extracted in this paper may not necessarily play a major role in trade. However, observing changes when breaking down the international trade network using complex network methods and relating these changes to other elements of international trade is a topic for future research. This aspect has been discussed in the discussion section. We added some sentences in line 510-519 and line 639-643.
Some smaller comments follow:
I was occasionally confused regarding the terminology, For example, it is not clear to me whether the terms “community” and “module” are synonymous. If so, it would be better to stick to one of them. The term “region” was used to refer to the years 1995-2005 and 2006-2018, respectively. Why not “time period”?
Reply: We have unified the terminology into “module”. We also change the term “region” into “time period”.
Line 81-82: It is stated that “Identifying the cores of communities and assessing their influence on various products can aid countries in selecting efficient and profitable trade partners.” How? Trade is typically determined by the competitiveness of firms – why would they be more interested in some partners than others? How are the active agents in international trade – the exporting companies – helped by this?
Reply:
This sentence is one that can lead to misunderstanding. The modular structure of a commodity trade network depicts the relationships between trading partners. These relationships are determined by the supply and demand for products between trading countries and are influenced by the global trade value chain. Trade networks can help you understand the relationships in these trade value chains and how they are affected by economic conditions. We modified the sentence in line 107-112.
Line 87-88: It is stated that the analysis helps to understand “how countries shift from their regional communities due to the optimization of the modularity function based on the Newman-Girvan algorithm”. What does this mean? Were the countries aware of these shifts, or did they occur inadvertently?
Reply:
Countries are not aware of module changes but can observe such changes in their trade networks. The sentence has been modified to convey the meaning clearly. Movements between modules in the trade network are measured using the Newman-Girvan algorithm, and these movements reflect the temporal variation in trade between each country. This movement did not occur by chance, but rather resulted from the interaction of the two countries through trade. We modified the sentence in line 116-117.
Line 124-125: The text argues that “Nodes with a central position in their clusters, meaning they have numerous links with other group partners, are likely to play essential roles in control and stability within the group”. It is not clear to me how this control would be exercised.
Reply:
The sentence has been modified. The hub nodes of a module play an important role in the network, and their importance is expressed in the network's connectivity. The roles of modules and hubs can be interpreted depending on the type of data that makes up the network. We modified the sentence in line 155-158.
I am bit confused by the meaning of “core position” in the community. Terms like “rule”, “govern”, “dominate”, “influence”, and “take charge” are used throughout the paper. What does this mean? How does a core country “rule”? Is it through its influence in regional trade agreements, or more covertly, through grand strategies and unilateral actions to maximize its soft or hard power?
Reply:
The core position refers to the node with the most connection in each module and corresponds to the local hub in the module. The local hub changes dynamically according to changes in trade. This change means a change in trade relations, and a change in the local hub means a large change in trade relations. Countries connected to a local hup form a strong trade correlation.
Are the vertical axes in Figures 3 and 4 defined the same way? It is confusing that the scales are so different, centered around 0.3 in Figure 3 and up to 50 times larger in Figure 4. An explanation is warranted.
Reply:
In Figure 3 the vertical axe represents the value of modularity defined in Eq. (1). However, in Figure 4 the vertical axe is the change of the modularity defined in Eq. (2).
Line 535-536: What is the meaning of the sentence “The food, mineral fuels, manufactured goods, and machinery equipment markets demonstrate a greater influence scale.”
Reply:
Our description is unclear. We delete this sentence.

Reviewer 3 Report
Comments and Suggestions for Authors
Dear Authors,
the following points will be provided:
- Title:
The title is good.
- Abstract:
The abstract needs to improved, please explain about the methodology of research more.
- Keywords:
I suggest more Keywords cab=n be written for this article.
- Introduction:
In the introduction it is necessary to address the novelty of the research at the end of introduction and it should be emphasized from different aspects.
Also explicitly specify what is the problem statement?
- Methodology:
What was the kind of research method in this study. Please explain.
- Results:
The authors can explain this part briefly.
- Conclusion:
Please mention the research limitations at the end of this section.
What is the main effect of the study?
Comments on the Quality of English LanguageMinor editing of English language required
Author Response
Reply to third reviewer.
Thank you for your fruitful comments.
the following points will be provided:
- Title:
The title is good.
- Abstract:
The abstract needs to improved, please explain about the methodology of research more.
Reply: We improved the abstract. We employed Louvain modularity optimization method which is explained in section 2.1
- Keywords:
I suggest more Keywords cab=n be written for this article.
Reply: We add more keywords.
- Introduction:
In the introduction it is necessary to address the novelty of the research at the end of introduction and it should be emphasized from different aspects.
Reply: We emphasized novelties and different aspects in the Introduction in line 107-112.
Also explicitly specify what is the problem statement?
Reply: We discuss the problem statement in last parts (last three paragraphs in line 113-130) of the Introduction section.
- Methodology:
What was the kind of research method in this study. Please explain.
Reply: We explain the research method in section 2.1-2.3. We applied the Louvain modularity optimization method which is explained in section 2.1.
- Results:
The authors can explain this part briefly.
Reply: Our results include many findings. It is very difficult to explain briefly. We keep the original form of the results part.
- Conclusion:
Please mention the research limitations at the end of this section.
Reply: We add limitation at the end of the Conclusion section.
What is the main effect of the study?
Reply: We discussed the main effect of the study in discussion part. We identify the module structure of the international trade network. The analysis of module dynamics provides valuable insights into global trade trends, fostering sustainability in trade practices, and comprehending the impacts of crises on various commodities.
Comments on the Quality of English Language
Minor editing of English language require
Reply: We modified English grammar and corrected typos.

Round 2
Reviewer 2 Report
Comments and Suggestions for Authors
The revision has improved the manuscript, but there are still a few weaknesses that need to be addressed. I will only note three issues here:
First, a language revision is necessary. Following my earlier suggestion, the authors are now systematically using the term "module", but it would be appropriate to carefully check whether this term is always appropriate after the "search - replace" operation carried out by the authors. In some instances, it does not sound correct -- sometimes "modular" or "modularity" would be better, but there are places where neither of these seem perfect.
Second, I am concerned about the lack of weights in the construction of the network. It is more clear in this revision that the positions of countries are explained only by the number of links (or an "extensive margin") rather than the strength of these links (or an "intensive margin"). This makes it difficult to understand what a position at the "core" of the network means. A more explicit discussion is needed to clarify that an influential or dominant position is not necessarily linked to what is normally understood as power or influence in international economics. The analysis mainly focuses on the question "Who do you trade with?" rather than "How much do you trade?".
Third, I am still unsure about how the findings can be useful for strategy or policymaking. The authors note now that "The analysis of module dynamics provides valuable insights into global trade trends, fostering sustainability in trade practices, and comprehending the impacts of crises on various commodities" (line 120-121). I would like to see some explicit discussion in the concluding part about what these important insights are. Now, the conclusion mainly states the weaknesses of the approach, i.e. that it does not include information about trade volumes or changes in global value chains (lines 635-640). It is therefore important to justify the exercise with a brief summary of the insight that have been reached thanks to the model exercise.
Perhaps one way here would be to emphasize that the various crises discussed in the paper have had substantial effects on the selection of trade partners. The results do not reveal whether the changes in trade volumes have been small or large (my guess is that a weighted network is less volatile), but a reasonable hypothesis is that modularity increases during crises because more distant markets will be appear more risky.
Comments on the Quality of English LanguagePlease review the use of the term "module".
Author Response
Reply to the reviewer’s comments.
Reply: Thank you for your good comments.
The revision has improved the manuscript, but there are still a few weaknesses that need to be addressed. I will only note three issues here:
First, a language revision is necessary. Following my earlier suggestion, the authors are now systematically using the term "module", but it would be appropriate to carefully check whether this term is always appropriate after the "search - replace" operation carried out by the authors. In some instances, it does not sound correct -- sometimes "modular" or "modularity" would be better, but there are places where neither of these seem perfect.
Reply: We check and modify the terminology. We replace with appropriate terminology related to "modular" or "modularity.
Second, I am concerned about the lack of weights in the construction of the network. It is more clear in this revision that the positions of countries are explained only by the number of links (or an "extensive margin") rather than the strength of these links (or an "intensive margin"). This makes it difficult to understand what a position at the "core" of the network means. A more explicit discussion is needed to clarify that an influential or dominant position is not necessarily linked to what is normally understood as power or influence in international economics. The analysis mainly focuses on the question "Who do you trade with?" rather than "How much do you trade?".
Reply:
The node with the core position is at the center of connectivity within the module. The node in the core position is similar to the local degree centrality node within the local module since it has the highest degree. While degree centrality nodes are defined in the entire network, core position nodes are defined within the module. Therefore, countries within the module have shorter paths connected to core position nodes, reflecting the efficient flow of trade between core position nodes and other nodes within the module in the network.
Most of the core positions measured were the USA, Germany, China, and Japan, which are central countries in world trade. Therefore, most countries connected to core position nodes have active trade relationships with each other. After checking the modules and core positions, we looked at the trade volume between the two countries and found that the trade volume between the two countries was large. Of course, countries with small trade volume, such as African countries, also have a strong tendency to be connected to the core position.
Third, I am still unsure about how the findings can be useful for strategy or policymaking. The authors note now that "The analysis of module dynamics provides valuable insights into global trade trends, fostering sustainability in trade practices, and comprehending the impacts of crises on various commodities" (line 120-121). I would like to see some explicit discussion in the concluding part about what these important insights are. Now, the conclusion mainly states the weaknesses of the approach, i.e. that it does not include information about trade volumes or changes in global value chains (lines 635-640). It is therefore important to justify the exercise with a brief summary of the insight that have been reached thanks to the model exercise.
Reply:
The modular structure extracted from the world trade network was observed to change depending on the economic environment. Changes in modularity could be observed before and after the economic crisis. Meanwhile, the modularity fluctuations of “Animal and vegetable oils, fats and waxes,” “Manufactured goods chiefly classified by material,” and “Food and live animals chiefly for food” fluctuate relatively small over the entire period. These items maintain stable trade relations. On the other hand, items excluding these were highly volatile. For highly volatile items, changes in trade relations intensified due to the economic environment and economic crisis. Linking these changes to trade stability and sustainability is left for further work.
Perhaps one way here would be to emphasize that the various crises discussed in the paper have had substantial effects on the selection of trade partners. The results do not reveal whether the changes in trade volumes have been small or large (my guess is that a weighted network is less volatile), but a reasonable hypothesis is that modularity increases during crises because more distant markets will be appear more risky.
Reply:
I agree your suggestion. However, we didn’t support that hypothesis in this work concretely. This assumption is left for further research in the future.